# Optimizing Access to the COVID-19 Vaccination for People Experiencing Homelessness

**DOI:** 10.3390/ijerph192315686

**Published:** 2022-11-25

**Authors:** Jane Currie, Olivia Hollingdrake, Elizabeth Grech, Georgia McEnroe, Lucy McWilliams, Dominic Le Lievre

**Affiliations:** 1School of Nursing, Queensland University of Technology, Kelvin Grove, Brisbane 4059, Australia; 2Homeless Health Service, St Vincent’s Hospital Sydney, 390 Victoria Street, Darlinghurst, Sydney 2010, Australia

**Keywords:** homelessness, vaccination, COVID-19, vaccination hesitancy, access to care, model of care

## Abstract

The success of the Australian COVID-19 vaccination strategy rested on access to primary healthcare. People experiencing or at risk of homelessness are less likely to access primary healthcare services. Therefore, leaders in homeless health service delivery in Sydney identified the need to develop a vaccine hub specifically for this vulnerable population. The aim of this study was to develop an evidenced based model of care to underpin the Vaccine Hub and optimize access to vaccination for people experiencing or at risk of homelessness. A mixed methods study was conducted that included interviews with key stakeholders involved in establishing and delivering the Inner City COVID-19 Vaccine Hub, and a survey with people receiving COVID-19 vaccination. Over the 6-month period of this study, 4305 COVID-19 vaccinations were administered. Participants receiving vaccination reported feeling safe in the Vaccine Hub and would recommend it to others. Stakeholders paid tribute to the collective teamwork of the Vaccine Hub, the collaboration between services, the ‘no wrong door’ approach to increasing access and the joy of being able to support such a vulnerable population in challenging times. The study findings have been populated into a Vaccination Hub Blueprint document that can be used as a template for others to improve access to vaccinations for vulnerable populations.

## 1. Introduction

People experiencing homelessness face challenges accessing vaccinations. The urgency of the Australian COVID-19 national vaccination program requires an ongoing, integrated, and collaborative approach to ensure that people experiencing homelessness are supported to access COVID-19 vaccination in a culturally appropriate way that fosters a sense of safety by acknowledging people’s sense of cultural identity. As of 30 July 2022, 20,160,781 (78.3%) people aged 16 years and over in Australia have received at least one COVID vaccine dose, and 19,817,623 (77.0%) have received two doses, and 14,095,910 (54.85) have received three [1]. Once vaccination became available, the Australian Government developed a three-phase national roll-out strategy [2]. In Phase 1a the strategy focused on frontline health workers, aged care and disability staff and residents, and quarantine and border workers. Thereafter, elderly adults, other health workers, Aboriginal and Torres Strait Islander people (The Torres Strait Islands stretch between the coast of Northern Australia and Papua New Guinea, with Indigenous populations that are distinct from those of Australia’s mainland) and then reaching adults aged 60–69 and the balance of the population. The final phase focused on those under the age of 18 years old.

In response to the national rollout, leaders in the provision of homeless health services became concerned that those experiencing homelessness may be at greater risk of exposure to COVID-19 due to their inability to self-isolate in a dwelling. Plans were made to establish a vaccine hub specifically focused to people experiencing homelessness. In this paper we report the findings of a study that explored the model of care underpinning the homeless health Inner City COVID-19 Vaccine Hub in Sydney Australia. These findings can be used as a template to assist other services to optimize access to vaccinations for people experiencing homelessness.

### 1.1. Homelessness in Australia

Homelessness is prevalent across Australia, with greater than 116,000 people experiencing homelessness [3]. The Bi-Annual City of Sydney Street Count (February 2022) reports 225 people sleeping on the streets in the Sydney Local Government Area and 269 people sleeping in crisis and temporary accommodation [4]. The Australian Bureau of Statistics’ (ABS) definition of homelessness describes a person as homeless if they do not have suitable accommodation, live in an inadequate dwelling, have no or limited/non-extendable tenure, or have no control of/no access to or space for social relations [3]. Homelessness is an umbrella term used to describe four broad population groups (1) Rough sleeping; (2) Supported accommodations (e.g., refuges & crisis accommodation; (3) Short –term accommodation without tenure (e.g., boarding houses, hostel, caravan, couch surfing); and (4) Accommodation in institutional settings (e.g., hospitals, drug and alcohol rehabilitation centers, jail) [5]. Aboriginal and Torres Strait Islander people are over-represented in the number of people counted as homeless in Australia, making up 3% of the total population but 20% of all persons who were homeless on Census night in 2016 [3]. Issues that contribute to homelessness, or escalate during a period of homelessness, are disproportionately experienced by Aboriginal and Torres Strait Islander people, including chronic disease, mental health concerns, social disadvantage, domestic violence and long standing inequity in access to health and social services [6].

### 1.2. Impact of Homelessness on Health

As one of the social determinants of health, access to stable accommodation has a profound impact on individuals’ health and wellbeing. The evidence linking homelessness and unstable accommodation with health outcomes is compelling. A 15-year retrospective longitudinal cohort study reported that one episode of any level of homelessness increased the risk of death over 15 years [7]. Homeless individuals had younger median age at death (66.60 vs. 78.19 years) and significantly increased mortality risk ratios compared to the non-homeless individuals [7]. Living on the streets or in unstable accommodation places people at greater risk of accidental injury and violence, musculoskeletal, skin, respiratory problems, poor oral health, drug and alcohol use and mental health issues [8,9,10,11,12].

Despite experiencing substantial health issues, people experiencing homelessness are less likely to access preventative, General Practitioner and/or primary health services, compared to those living in stable accommodation. Instead, this vulnerable population are more likely to seek healthcare from the emergency department, often at a later stage of ill health [13]. When attending an emergency department, they often wait longer to receive care [14], leave before being seen by a health professional, and re-present at a later time [15,16,17] compared to those in stable accommodation. This fragmented approach to healthcare access can result in costly hospital admissions and the development of chronic illness [5,13].

### 1.3. Approach to COVID-19 Vaccination

The success of the Australian COVID-19 vaccination program required that most of the Australian population become immunized. To achieve this, it was important that all Australians had access to the COVID-19 vaccination. The Australian Government led the delivery of the COVID-19 vaccination program through primary healthcare by providing General Practitioners (GP) specific incentives and subsidies to prescribe and provide the vaccination program. This approach was successful for many who were willing and able to access primary healthcare services. Leaders in Homeless Health in Sydney anticipated that people experiencing homelessness would struggle to access vaccination through primary healthcare services and therefore established a vaccine hub specifically for people experiencing homelessness. 

### 1.4. The Vaccination Hub Matthew Talbot Hostel and St Vincent’s Hospital Sydney

The Inner City COVID-19 Vaccine Hub was established in May 2021, to improve access to vaccination for people sleeping rough, people in specialist homelessness services and people at risk of homelessness, such as those living in social housing or temporary accommodation. The Vaccine Hub was initiated by stakeholders at St Vincent’s Hospital Sydney, South Eastern Sydney Local Health District and the Kirketon Road Centre. Engagement was sought from health and non-health partners within inner city Sydney to establish a collaborative approach. A process for the sharing of resources, such as Accredited Nurse Immunizers, consent forms and a common approach to messaging was established. The Vaccine Hub was promoted using flyers, which were distributed through existing networks. Stakeholders committed to the Vaccine Hub included St Vincent’s Health Network, South Eastern Sydney Local Heath District, Kirketon Road Centre, Sydney Local Health District, City of Sydney Council, Department of Communities and Justice, Neami National and St Vincent DePaul Society including the Matthew Talbot Hostel. Many of these organizations were members of the Intersectoral Homelessness Health Alliance, designed to take a community wide collaborative approach to improving access to healthcare and health outcomes for people experiencing or at risk of homelessness in Sydney. The first Vaccine Hub was implemented on Thursday 20 May 2021 at the Ozanam Learning Centre next to the Matthew Talbot Hostel, Woolloomooloo and has run weekly since. Over the 6-month period of this study, 4305 COVID-19 vaccinations were administered through the Vaccine Hub.

The overall purpose of this research is to develop an evidence-based best practice model of care to underpin the Inner City COVID-19 Vaccine Hub in order to optimize access to vaccination for people experiencing or at risk of homelessness. It is intended that the findings of this study will be rapidly translated into practice to underpin the development of the existing Vaccine Hub and be used as a best practice model of care for other geographical locations. 

### 1.5. Aims and Objectives

The primary objective of this study was to develop a best practice model of care to optimize access to the COVID-19 vaccination for people experiencing homelessness in inner-city Sydney. The secondary objectives included: Documenting the model of care used at the Inner City COVID-19 Vaccine Hub.Exploration of the perceptions of key stakeholders involved in delivering and designing the Inner City COVID-19 Vaccine Hub model of care in relation to its effectiveness at improving access to vaccination, andExploration of the perceptions of client’s receiving the COVID-19 vaccination regarding the Inner City COVID-19 Vaccine Hub model of care and its impact on their access to vaccination, including vaccine hesitancy.

## 2. Materials and Methods

A mixed method study was used. The study used semi-structured interviews with key stakeholders to document the Inner City COVID-19 Vaccine Hub (Vaccine Hub) model of care and a survey was used to explore the perceptions of people receiving the COVID-19 vaccination at the Vaccine Hub on their access to vaccination. 

### 2.1. Interviews

Semi-structured interviews were conducted with a purposive sample of stakeholders involved in the establishment and delivery of the Vaccine Hub. Fifteen participants were invited by email to participate, and 12 interviews were conducted. The roles of participants were varied and included health professionals, health service managers and those in non-health managerial and executive roles. Each interview was conducted by the lead author (JC) and lasted for between 20.03 to 48.02 min. The interviews were conducted using the video conference platforms, Zoom [18] or Teams [19] (Microsoft Teams). Each interview was recorded with participant permission and professionally transcribed. An interview schedule was designed by the authors and piloted with people with lived experience of homelessness, and clinicians and managers in the Homeless Health Service at St Vincent’s Hospital Sydney. The interview schedule (Table 1) was further piloted on the first two interviews undertaken and as no changes were required, these interviews were included in the final sample of data. 

### 2.2. Survey

A survey was conducted with a convenience sample of participants who consented to participate following the receipt of their COVID-19 vaccination. Eligible participants were those aged over 18 years and were able to provide informed consent to participate. Participants were approached by a registered nurse member of the research team during the 15 min waiting period, post receipt of their COVID-19 vaccination in the Vaccine Hub. The research team member was accompanied by a Peer Support Worker with lived experience of homelessness and/or an Aboriginal Health Worker. The purpose of the research was explained to each participant and a participant information sheet was provided, which was read to participants as required. Participants were invited to ask any questions. Consent to participate was provided verbally and was electronically recorded on the survey tool by the research team member. The survey was verbally administered to participants by the research team member who recorded the responses via the REDCap survey tool using a mobile device. The survey was designed by the authors, to explore the perceptions of clients in relation to whether the Vaccine Hub improved their access to the vaccination. The survey was piloted on six members of the Homeless Health Service, St Vincent’s Hospital Sydney, including one Aboriginal Support Worker and a Peer Support Worker with lived experience of homelessness, two nursing clinicians and a health service manager and a mental health clinician. The survey was conducted between 23 September and 28 October 2021 at each weekly Vaccine Hub session. The research team member collecting the data was sometimes required to support the administration of vaccines and was available to administer the survey for half day periods only. The frequency of data collection is reflected in the number of surveys completed (*n* = 49). 

### 2.3. Data Analysis

The qualitative data were analyzed inductively using the Braun and Clarke approach to analysis [20]. Two authors independently read (familiarization) the transcripts (OH, LG) and then met to code the data together (coding). Once all transcripts were coded the authors met again to theme the codes (theming). A third author (JC) deductively analyzed the data to identify any specific learnings to enhance the initial draft of the Vaccination Blueprint (Appendix A). Quantitative data was exported into SPSS (IBM V27) and analyzed by one author (GM) using descriptive statistics. 

## 3. Results

The results relating to the consumer survey of people accessing the Vaccine Hub are presented first, followed by the qualitative findings from the stakeholder interviews. The study recruited a total of 49 participants following their COVID-19 vaccination at Inner City COVID-19 Vaccine Hub. Majority of participants identified as male (73.5%) with a median age of 43.4 years (SD = 16.3). Approximately 79.6% of participants disclosed as non-indigenous (neither Aboriginal nor Torres Strait Islander people), and the most frequently reported sleeping accommodation by participants were either a house/apartment or social and community housing (Table 2).

All study participants reported feeling comfortable receiving their vaccination at the Inner City COVID-19 Vaccine Hub, with most participants attributing their comfort to a positive experience with staff (71.0%, *n* = 22), and the Hub being a safe environment (22.6%, *n* = 7). Free text comments suggested that participants had confidence in the ability of those providing care in the Vaccine Hub *“…everyone seemed that they knew what they were doing*”, it was a “*clear process and was relatively quick*” and welcoming “*friendly, nice atmosphere*”. Nearly one third (31.3%, *n* = 15) of respondents identified personal health concerns as a primary motivation for receiving the COVID-19 vaccination, followed by community health concerns (16.7%, *n* = 8) and a general desire to return to ‘normality’ (16.7%, *n* = 8). Following their vaccination, 91.8% (*n* = 45) of participants reported that they would recommend the COVID-19 vaccination to others. 

Most participants (69.4%, *n* = 34) reported knowing someone who does not want a COVID-19 vaccination. Participants reported that others fear vaccine-related side effects (25.0%, *n* = 8), they are misinformed on the vaccine (21.9%, *n* = 7), they do not believe COVID-19 exists (15.6%, *n* = 5), and they are unaware of the long-term effects of the vaccine (15.6%, *n* = 5). In response to strategies that may encourage vaccination in those reluctant, participants largely recommended reiterating that they and others have received the COVID-19 vaccination and are unharmed (22.6%) and that most people are unlikely to suffer any serious side effects due to COVID-19 vaccination (16.1%). Almost 10% (9.7%) of respondents reported that they believed no strategies will encourage COVID-19 vaccination in those known to them that are reluctant.

Personal participant hesitancy (slightly hesitant, quite hesitant, extremely hesitant) towards vaccination was reported in 61.3% (*n* = 30) of cases, and the leading reasons for hesitancy were misinformation regarding COVID-19/COVID-19 vaccine (30.3%, *n* = 10) and skepticism surrounding the development of the COVID-19 vaccine (27.3%, *n* = 9). Free text comments indicated many participants had a fear of receiving an injection, “*mild to extreme needle phobia*” “*Don’t like needles*”. Chi square analysis revealed no significant associations between COVID-19 Vaccine Hub participant survey responses and participant demographic characteristics, including age, gender, and indigenous status (Table 3).

### 3.1. Interviews

Analysis of interview data is reported through five broad themes, (1) access to care (2) knowing community (3) person-centered practice (4) team strengths and (5) model of care. Each of the participants demonstrated a genuine enthusiasm for their participation in the Vaccine Hub. The overriding sentiment was one of wanting to help an extremely vulnerable population during a time of great uncertainty and potential risk. Each theme is discussed briefly below, the specific codes and themes are shown in Table 4.

#### 3.1.1. Theme 1: Access to Care

Motivation for stakeholders’ involvement in the Vaccine Hub was their shared passion to increase access to care. Stakeholders understood that those experiencing or at risk of homelessness would struggle to access vaccinations using traditional services and expressed a strong conviction that all people should be afforded this right. As one stakeholder stated, “*If you can say access is blocked to the general population, then it’s definitely blocked to the homeless population*” (Participant 2). Stakeholders understood that ensuring access to the COVID-19 vaccination for marginalized people required a different approach,
“*How are we going to get rough sleepers to GPs? To clinics? To appointments?”, it’s impossible. It’s a barrier, and it’s going to be too difficult…you bring a service to them, or in a location that we know is accessible and where there are a lot of rough sleepers, it’s much easier for them to access it.*” (Participant 6)

The organization and establishment of the Vaccine Hub involved multiple health, social services and community organizations including the City of Sydney Council. By joining forces around a common goal, these stakeholders perceived that they improved access to care swiftly*, “All the managers…could see very quickly and clearly that this was a no-brainer” (Participant 2). At the time of beginning the Vaccine Hub there was genuine uncertainty around the eligibility for vaccination, as one participant remarked,’…to be able to bring this to the population that usually is last in line. Make them first in line.”* (Participant 7).

Stakeholders understood the need to prioritize vaccination for homeless and socially marginalized people, many of whom have competing priorities impacting their ability to adhere to public health orders and exercise physical distancing. As a consequence of their marginalization and conditions of living, this population experience increased rates of physical illness, mental illness and substance use disorders when compared to the general population. 

“*Our clients are really vulnerable, and they have a whole range of comorbidities, and it was very important to move very quickly and get them vaccinated as quickly as possible in the most efficient, effective way.*” (Participant 8)

#### 3.1.2. Theme 2: Knowing Community

Trust within the community was a key driver of success for the Vaccine Hub. Many of the organizations involved had pre-established relationships and services with each other and with service users. Stakeholders leveraged these relationships to promote and facilitate vaccination “…*the trust that we had in the community, was probably how we got people to come”.* (Participant 12). Stakeholders expressed a desire to create a space where people would feel safe and valued and where meeting the needs of each individual was prioritized by “*…recognizing the importance of relationships*”. (Participant 5).

Stakeholders strove to make the Vaccine Hub a place of social connection and unity during a time of fear, mistrust, and loneliness. “*Clients in particular found COVID so isolating, and so lonely*.” (Participant 6). Stakeholders emphasized the importance of creating a culturally safe place for people experiencing or at risk of homelessness, and that more could have been done to increase accessibility for culturally and linguistically diverse populations. This included more interpreters, information in different languages and greater involvement of First Nations staff and community members in the daily running of the service. One stakeholder reflected,
“*We did the best that we can, but—I mean, I think it’s just us as services, aren’t we, where, as government services, we’re set up for white people, for white Australians.*”(Participant 4)

The Vaccine Hub staff wore T-shirts with Aboriginal symbology to improve the approachability of the service for First Nations people and ‘…*to make Aboriginal and Torres Strait Islanders feel that they could ask us to find them someone culturally appropriate to speak to*…’ (Participant 1). There were Aboriginal Health Workers and Peer Support Workers in the Vaccine Hub, and they were critical because “…*one of the most important things for Aboriginal people to talk about vaccination is the Aboriginal Health Workers and Aboriginal Elders*.” (Participant 4). The stakeholders were able to facilitate access to vaccination for refugees by providing “…*a safer space for them to not wait in the crowd because that was going to be triggering for them*.” (Participant 10). Universally, stakeholders recognized that person-centered and trauma-informed practices were essential to meet community needs, compelling them to seek adaptable and collaborative ways of working.

#### 3.1.3. Theme 3: Person-Centred Practice

Stakeholders stated the Vaccine Hub upheld a “no wrong door” standpoint, which enabled a flexible and tailored person-centered approach that optimized accessibility. No wrong door meant the stakeholders made every effort to accommodate the specific needs of people seeking vaccination. For example, some people were hastened through the waiting process to enable their access, “*The people that couldn’t wait…that person obviously has some psychosis, and we just need to shift them through.*” (Participant 7). Another example, “*…there was one gentleman he struggled, he really just struggled to follow instructions…I think if we didn’t have a good understanding that his cognitive capacity was limited, yeah, somebody like that would’ve been turned away from a different hub.*” (Participant 8).

There was a desire to provide evidence-based information and education without coercion, which meant allowing opportunities to discuss vaccine hesitancy, dispel myths and answer questions, even if this took a substantial amount of time. 

“*That is a problem in healthcare, I suppose, is the messages change an awful lot. We were just saying, “You can come in. This is really good. This is why you should have it. We’ll be available. Don’t worry if you don’t want it. We’ll have it next week.*” (Participant 2)

Because the Vaccine Hub was situated in a fixed location, promotion and outreach were needed to ensure community members did not miss out. Stakeholders were able to engage people they had an established relationship with and encourage them to attend. “*[Staff member] had a man living in a cave down in the national park that he brought in to get jabbed…*” (Participant 1). As the weeks progressed, staff deployed to streets and public housing sites to reach those struggling to visit the Vaccine Hub. The mobility of the service allowed an opportunity for some of the harder to reach populations to be vaccinated. 

“*They’ve been vaccinating people in all sorts of places. There was a couple of Nepalese students who got displaced because of the uni shutdown and we were able to jab them down where they were sleeping down at [park].*”(Participant 1)

The Vaccine Hub provided a platform for more interventions than just vaccination, acknowledging that people experiencing homelessness are less likely to access primary and preventative health care than people in stable accommodation. Examples included on-site Hepatitis C testing, addressing housing needs, and provision of food. 

“*We had people asking about emergency housing. We had women in crisis…and you had all these people from other services standing there guiding the way, and they had the knowledge right there and they could do it subtly in a really streamlined way.*” (Participant 9)

#### 3.1.4. Theme 4: Team Strengths

Teamwork and collaboration were essential to the Vaccine Hub’s success. Stakeholders reflected on the high quality of leadership demonstrated from the outset and the important contributions of individuals. The shared goal of promoting vaccine access was critical. 

“*I guess it’s like teamwork and collaboration. But I think there was a willingness—we all had one single mind which was really to protect our community, to protect vulnerable people, to support people who we knew found it more difficult to access healthcare.*”(Participant 4)

Stakeholders conveyed a sense of goodwill and camaraderie surrounding the Vaccine Hub. They shared numerous examples of people’s willingness to contribute their time and skills or networks, to ensure its smooth running. Stakeholders expressed genuine enjoyment and fulfillment in the work. Many staff took time out of their routine work duties to take part in the Vaccine Hub, others volunteered their time for free, “*…people were so committed and so loved the project that they were just there…it has shone out in this project.*” (Participant 2). Courageous and enterprising leadership from key team members provided the impetus to establish the Vaccine Hub, and to put ideas into practice in the context of a looming health crisis. As one participant stated, “*if you insist on having a perfect project in place before you start, people will die*” (Participant 2). In combining the skills and resources of many individuals and services there was a notion of no egos being involved; everyone was working towards a common goal and every role was important, “*It just made more sense to work together, rather than siloed and working in isolation.*” (Participant 9).

The Hub relied on inter-agency collaboration, including some organizations and professionals that have not previously worked closely together. For example, St Vincent’s Homeless Health service provided the vaccinations and were instrumental in the initiation and running of the service as were South Eastern Sydney Local Health District, and the Kirketon Road Centre, and the Ozanam Centre provided a premises. The Matthew Talbot Hostel health service provided clinical support and the City of Sydney helped with food and logistics. 

“*There were no egos around which service should be doing what, who should be doing this. And you have really senior people standing outside and making sure that people are filling out consent forms.*”(Participant 12)

Another driver of success was the perseverance of the staff involved. Working through challenges such as vaccine misinformation, long queues, physical distancing, changing regulations, limited vaccine supply and vaccine hesitancy in the community, the team adapted their practice and continued working together toward the shared goal. 

“*It was definitely learning on the fly. And I think also… we were presented with issues that we hadn’t anticipated…and that was something we had to adapt to and learn from as well.*”(Participant 12)

Perseverance proved worthwhile, as momentum took over and the Vaccine Hub gained popularity within the community. People who received a vaccine began speaking with others about their experience, and access to vaccination began to flourish. 

“*It’s that word of mouth, so one person gets vaccinated and then they’re okay, so then their family members get vaccinated. And so, the whole family are vaccinated now… it was like dominoes.*” (Participant 4)

#### 3.1.5. Theme 5: Model of Care

The model of care for the Vaccine Hub (Appendix A) centered around the common goal to increase access to COVID-19 vaccination for people experiencing or at risk of homelessness. Stakeholders aimed to create a model that was replicable and could be used again for other public health initiatives. Innovative processes and partnerships were integral to achieving this aim, and the Vaccine Hub was expanded to cover new geographical locations
“*In partnership still with similar partners and some extra ones, [we] set up the Northcott Hub and the Lexington Place Hub—same models, which have equally done a huge job around those populations, particularly moving more into the social housing; lots of people who are also experiencing homelessness there. So, I guess one of the things of whether a model works is, is it duplicatable, replicable elsewhere, and it certainly has been.*” (Participant 5)

Cognizant of the impact of stigma and discrimination when accessing healthcare, stakeholders strove to create a setting where people could have their needs met by staff they trusted. Collaboration was integral to this model and stakeholders commented on the ways the Vaccine Hub differed from other projects by changing the way organizations interacted. Ensuring the most vulnerable community members were prioritized was an important aspect of the model. Over the course of the Vaccine Hub’s operation, efforts were hampered by misinformation, distrust and fear regarding COVID-19 and the vaccinations. Navigating these challenges was an integral aspect of the model of care. Staff needed to adopt flexible working patterns to manage community members’ concerns and ensure non-judgmental interactions. 

“*Every week, there was somebody in tears because they were so anxious around the vaccine. People came in and then changed their mind, and then they’d want to talk to someone…The anxiety around it for a lot of our clients, and the paranoia for some of our clients as well, was huge and immense... I think that’s also something that I just really loved that space. Being allowed, being able, to provide people that space and the time, and what they need.*” (Participant 8)

## 4. Discussion

These findings suggest that whilst vaccine hesitancy existed amongst two thirds of people accessing the Inner City COVID-19 Vaccine Hub, all participants reported feeling comfortable in the Vaccine Hub and most perceived it as a positive environment and would recommend the vaccine to others. These findings resonate with the intent of establishing the Vaccine Hub, which was to collaborate around the common goal of optimizing access to COVID-19 vaccination through a person-centered and trauma informed approach. As far as we know this is the first study to explore COVID-19 vaccine hesitancy among people experiencing or at risk of homelessness in Australia. 

The World Health Organization has defined vaccine hesitancy as “…the delay in acceptance or refusal of vaccines despite availability of vaccine services” [21]. There are many reasons for vaccination hesitancy and these have been summarized through the 5 C’s model of vaccination hesitancy [22] as follows: confidence (trust in the safety of the vaccine and the system delivering it, including motivations of policy decision makers), complacency (not perceiving disease as high risk), constraints (structural and psychological barriers), calculation (engagement in extensive information searching), and collective responsibility (willingness to protect the community through vaccination). Each of these 5 C’s are evident in the responses from participants receiving vaccination at the Inner City COVID-19 Vaccine Hub. The most prolific reason underpinning hesitancy was low confidence in the vaccine itself and suspicion of the policy makers who insisted on COVID-19 vaccination for public health safety. This form of hesitancy was challenging to overcome for those delivering the Vaccine Hub. Stakeholder accounts suggest that providing information through a tailored approach where people had time to consider and think about whether they wanted to receive the vaccination was critical to ensuring equity in access for people experiencing or at risk of homelessness. 

Once COVID-19 vaccination became available, it was acknowledged that demand would far outstrip supply and therefore certain groups were prioritized to receive vaccination [23]. Interview participants spoke of their genuine excitement at being able to prioritize those experiencing homelessness for vaccination, a group that are usually last in line, were now first in line. There was grave concern that people experiencing homelessness in Australia were at higher risk of contracting COVID-19 [24] and would have poorer outcomes if they did [25].

One of the strengths of the Inner City COVID-19 Vaccine Hub was its person-centered, trauma informed approach tailored to the homeless community. Being able to leverage existing relationships with the community and the service providers appeared to improve uptake of the COVID-19 vaccine. While the Inner City COVID-19 Vaccine Hub was provided through a fixed location, the model of care did evolve and expand to include outreach, in which people were offered transport to the Vaccine Hub or were provided the opportunity to receive vaccination in their residence or another location. It was outside of the scope of this study to include any evaluation or comparison of these outreach services. Therefore, we cannot provide any indication of whether a fixed clinic location was more effective than a mobile outreach approach or whether combining the two approaches would have been optimum. 

The UK has several successful examples of tailored approaches that ‘take the vaccine to the person’ [26]. These mobile outreach services developed in response to an understanding that many people did not wish vaccination in a mass clinic or in a National Health Service facility [23]. Instead, vaccinations were provided in less formal spaces such as refuges and soup kitchens, during and outside of usual business hours to optimize access to those who have different routines [26]. The Maximizing Uptake Programme in the United Kingdom, targeted towards people experiencing homelessness and other marginalized populations led to administration of almost 8000 COVID-19 vaccinations in a 7-month period [27]. Success was attributed to outreach work coupled with a targeted engagement and communication campaign that had been co-designed with community stakeholders. 

The Centre for Disease Control facilitated mobile COVID-19 vaccination services by providing resources to support the set up and service delivery of mobile clinics designed to reach underserved, high risk groups [28]. An example of a tailored and targeted approach to COVID-19 vaccination in Tennessee, United States, is the Meharry Medical College Mobile Vaccination Program. The Mobile Vaccination program partnered with other community organizations, including faith-based organizations and a School of Nursing to provide free COVID-19 vaccinations to targeted, underserved communities [29]. Together, these targeted services held the same goals as those involved in the Inner City COVID-19 Vaccine Hub, to provide equity in access to vaccination among those at highest risk of acquiring COVID-19 and those likely to be more profoundly impacted by succumbing to COVID-19. Our findings along with those of others do suggest that a flexible approach is critical to meet the emerging needs of populations during the COVID-19 public health crisis response [30]. 

The findings of this study have been used to underpin a model of care for the Inner City COVID-19 Vaccine Hub called the Vaccination Hub Blueprint (Appendix A). It is hoped that the Vaccination Hub Blueprint, including its philosophy of care, collaborative practices, resources, logistics and lessons learned, will be helpful to other services that provide vaccination to people experiencing, or at risk of homelessness, and to those experiencing any form of social marginalization. 

### Limitations

This study has some limitations. The sample size (*n* = 49) of the survey participants is small and may not be representative of all those experiencing homelessness in Sydney. The survey was administered in the 15 minutes following vaccination. The timing of obtaining this data may have influenced the responses, and with greater time to reflect, participants may have provided alternative responses. Only people who received the vaccine were invited to complete the survey, which means the survey responses somewhat presupposed the success of the Vaccine Hub. The views of people choosing not to engage with the Vaccine Hub are absent from the findings. There may have been some valuable feedback from those who did not engage with the Vaccine Hub. Regarding the interviews, a purposive sample of stakeholders was obtained. Therefore, the perceptions and views of the interview participants may not be directly transferable to other sectors or contexts. 

## 5. Conclusions

Access to vaccination can be challenging for people experiencing or at risk of homelessness. In establishing the Inner City COVID-19 Vaccine Hub, the intent was to prioritize this vulnerable population to ensure they received equitable access to vaccination. People using the Vaccine Hub perceived feeling welcome and were comfortable in the environment, despite their hesitancy towards the COVID-19 vaccine. The stakeholders delivering the Vaccine Hub leveraged their existing relationships with the community and key organizations to ensure that a person-centered and trauma informed approach was achieved. The Inner City COVID-19 Vaccine Hub was an exemplar of teamwork, with multiple organizations combining forces towards a shared goal of protecting some of our most vulnerable members of the community. 

## Figures and Tables

**Table 1 ijerph-19-15686-t001:** Interview questions.

Questions
What is your role in the Vaccine Hub?
What was the inspiration for establishing the Vaccine Hub?
Which community partners are involved in the Vaccine Hub and what is their role?
Does the Vaccine Hub have a philosophy of care? Can you share your thoughts on what it is?
How was the model of care for the Vaccine Hub developed?
Do you think the Vaccine Hub is culturally appropriate for all cultures and the staff are skilled to provide culturally safe information and services?
How do patients know about the Vaccine Hub?
What is the flow of care through the Vaccine Hub?
Why do you think the Vaccine Hub has been so successful?
What do you think patients like about the Vaccine Hub?
What has worked well/less well in delivering the Vaccine Hub? What are the factors influencing this?
Have you noticed any hesitancy in the patient’s attending the Vaccine Hub, in relation to them receiving the COVID-19 vaccine? If so, are you able to elaborate as to why you believe people are hesitant?
What feedback, if any, do the patients accessing the Vaccine Hub provide to you?
Are there any improvements you would recommend to the Vaccine Hub design?

**Table 2 ijerph-19-15686-t002:** Sociodemographic characteristics (N = 49).

Sociodemographic Variable	Frequency *n* (%)
*Gender*	
Female	13 (26.5)
Male	36 (73.5)
*Age, mean [SD]*	43.4 [16.2]
*Age, years*	
18–24	6 (12.2)
25–34	8 (16.3)
35–44	10 (20.4)
45–54	12 (24.5)
55+	11 (22.4)
Not reported	2 (4.1)
*Indigenous status*	
Aboriginal	7 (14.3)
Torres Strait Islander	1 (2.0)
Both Aboriginal and Torres Strait Islander	1 (2.0)
Neither	39 (79.6)
Not reported	1 (2.0)
*Most frequent sleeping conditions*	
Streets (rough sleeping)	4 (8.2)
Crisis or emergency accommodation	4 (8.2)
Staying with friends or family	6 (12.2)
Social or community housing	9 (18.4)
Hostel	5 (10.2)
Boarding House	2 (4.1)
Rehabilitation facilities	3 (6.1)
House/Apartment	15 (30.6)
Not reported	1 (2.0)

Abbreviations: SD—standard deviation.

**Table 3 ijerph-19-15686-t003:** COVID-19 Vaccination Hub Survey questions and respondents (inner-city Sydney study population) answer frequencies and percentages (*n* = 49).

Question Possible Responses	Frequency *n* (%)
**What made you decide to get a vaccination today?**	** *48 (98.0)* **
It was a spontaneous decision	1 (2.1)
Someone told me it was important to get one	6 (12.5)
My friend/partner/someone I know was coming so I came along	1 (2.1)
I wanted to get one	6 (12.5)
The vaccination is free/easy to access	1 (2.1)
It’s important for my health and I don’t want COVID-19	15 (31.3)
It’s important for community health	8 (16.7)
Other	
Desire for normality (i.e., visit shops/work)	8 (16.7)
For travel purposes	2 (4.2)
Felt compelled/pressured to get vaccinated	3 (6.3)
To visit and protect loved ones	6 (12.5)
**Do you feel you have a good understanding of the vaccination process and potential side effects?**	** *49 (100)* **
Yes	44 (89.8)
No	5 (10.2)
**What would you like to know more about?**	5 (10.2)
Vaccine efficacy/vaccine effectiveness	1 (2.0)
Possible vaccination side effects	3 (6.1)
Information on Immunisation Certificate	0 (0.0)
Nothing	1 (2.0)
**Do you feel comfortable receiving your vaccination at the vaccine Hub?**	** *49 (100)* **
Yes	49 (100)
**Why did you feel comfortable receiving your vaccination at the vaccine Hub?**	** *31 (63.3)* **
Positive staff experience	22 (71.0)
Organised and quick delivery of service	6 (19.4)
Convenient location/easy access to service	5 (16.1)
Safe environment	7 (22.6)
**Now that you have received a COVID-19 vaccination yourself, would you recommend it to others?**	49 (100)
Yes	45 (91.8)
No	4 (8.2)
**If no, why would you not recommend it to others?**	***4* (100)**
It is their decision/their choice	4 (100)
**Do you know anyone who does not want to get vaccinated?**	*49 (100)*
Yes, I do	34 (69.4)
No, I don’t know anyone	15 (30.6)
**Do you know why they don’t want the COVID-19 vaccination?**	** *32 (65.3)* **
I don’t know why they don’t want the vaccine	3 (9.4)
They are scared of the side effects	8 (25.0)
They don’t believe COVID-19 exists	5 (15.6)
They simply don’t want the vaccine	2 (6.3)
They don’t know the long-term effects of the vaccine	5 (15.6)
They feel compelled/pressured to get the vaccine	1 (3.1)
They are misinformed on the vaccine	7 (21.9)
They are unmotivated to get the vaccine	1 (3.1)
Other	1 (3.1)
**If you knew someone who didn’t want the vaccine, what may encourage them to get vaccinated?**	** *31 (63.3)* **
Tell them that I and others have received the vaccine	7 (22.6)
Tell them it doesn’t hurt	3 (9.7)
Tell them it’s important to protect them from COVID-19	1 (3.1)
Tell them most people are unlikely to have serious side effects	5 (16.1)
Tell them the staff at the Vaccine Hub are friendly/organised	1 (3.2)
Provide them more information on COVID-19 vaccine	3 (9.7)
Tell them it’s important to protect the community and their loved ones from COVID-19	1 (3.2)
Enforce restrictions on the unvaccinated	3 (9.7)
Contracting COVID-19 and becoming ill themselves	3 (9.7)
Nothing will encourage them	3 (9.7)
Other	1 (3.2)
**How hesitant were you about receiving the COVID-19 vaccine?**	**49 (100)**
Not at all hesitant	19 (38.8)
Slightly hesitant	14 (28.6)
Quite hesitant	12 (24.5)
Extremely hesitant	4 (8.2)
**Why do you feel hesitant/not hesitant?**	** *33 (67.3)* **
**Not at all hesitant:**	
Understand the need/benefits of the COVID-19 vaccine	2 (6.1)
Felt safe in the Vaccine Hub	1 (3.0)
People I know have been vaccination and they’re okay	3 (9.1)
**Slightly—extremely hesitant:**	
Discomfort/fear towards needles and general vaccine symptoms	3 (9.1)
Individual medical reasons (e.g., history of clots)	3 (9.1)
Misinformation on COVID/COVID-19 vaccine	10 (30.3)
Sceptical of COVID-19 vaccine development	9 (27.3)
Other	2 (6.1)

Note—Respondents could select multiple answer choices hence percentages and totals in this table are based on respondents.

**Table 4 ijerph-19-15686-t004:** Qualitative interview themes.

Theme	Code	N=	Example
Access to care	Inspiration/ ethos	46	It’s about the marginalised and the people who have least. And there are few projects that I think would actually fit with the mission of all of these places so well. And so in that way, it was a no-brainer from the top down (P2)
	Vaccine availability	20	The fact that we had access to vaccines, and that was never questioned. Vulnerable people were prioritized, which is fabulous. (P9)
	Equity/ equality	11	They did a lot of work sort of vetting the people waiting in the queue and figuring out who was appropriate to attend the hub. (P3)
	COVID risk	7	The higher incidence of chronic disease and other things that would make them more susceptible to having severe COVID or getting really sick from COVID if they were to contract it. (P4)
	Desperation	3	It was interesting when you’ve got people from Vaucluse turning up in their Mercedes to a men’s homeless hostel to get vaccinated. I mean, that shows a certain level of desperation, doesn’t it? (P4)
Knowing community	Cultural safety	26	I think with homelessness, it’s all about access, but it’s also about doing it in a culturally appropriate way—as in culturally for homeless people, not culturally for a particular type of culture, of course—providing it in a trauma-informed patient/person-centred way. (P2)
	Rapport	10	Yeah, to just be in a quiet space with that. Because sometimes there’s a lot of people around, going back and forth, so sometimes people just need a bit of a quiet space with somebody that they trust just for five minutes, which is totally fine. (P8)
	Fostering social connection	4	Perhaps the normal doctor/patient barrier or something like that was a little bit lower. It was more welcoming. It was more fun, more teamwork than perhaps you would normally have when dealing with people. Then it was so enjoyable because you like to connect with people at that level, don’t you? (P7)
	Community involvement	1	You were seeing some of the residents and the clients, they were helping out in terms just getting things set up, stuff like getting the food ready and all that sort of stuff. I think there was that sense of ownership on that space as well. (P11)
Person- centred practice	Navigating vaccine hesitancy	21	I think because it didn’t feel like a clinical space as well, that’s probably the other thing. It was familiar, it was home, basically, for a lot of people. (P11)
	Outreach	15	Some of these children with severe disability, who were living in group homes down in the Shire, we were able to link in with them. There must have been 15 of these kids. (P1)
	No wrong door	12	We tried to take snacks along and lots of chats. There are a lot of people who said, “Oh, I had a great conversation.”
	Opportunities for intervention	9	Having access to other services whilst they were there, having Housing and Neami, some food. (P3)
	Advocacy	7	Our experience in working with that vulnerable population told us that flexible models of care are required to ensure access to services… The mainstream ways of delivering vaccine through the eligibility checker and all that was just going to be beyond the access of the people that we were talking about. (P12)
Team strengths	Partnership and collaborating	39	I’ve been in community services nearly 20 years and having seen the true collaboration… you don’t see very often… It didn’t feel like there were agendas being brought into the hub. People were really going, “How do we work together to make this happen?” (P10)
	Goodwill	19	I think COVID was disruptive for a lot of people in frontline services, and for them to be like, “We’re providing a solution. We’re not just feeling like we don’t know how to help other people.” It’s like, this is something really tangible that we can do. (P10)
	Perseverance and momentum	19	It already feels like we’re working together more collaboratively with other stuff, not just the hub. (P3)
	Passion and drive	11	You’ve got a group of people there… all kinds of workers, and they like their work and they like working with people. And so they provide. They’re happy and they’re open and they’re welcoming and they’re warm. And imagine if that happened in every health centre in Australia? (P2)
	Initiative	6	We basically put [retired GP] on the entrance, and so you would walk into the foyer and here she is, down on one knee, helping people fill in forms and talking them through any of their hesitancy, any of their fears or any of their concerns. (P1)
	Leadership	4	I suppose independently and outside of COVID, everybody is scratching for money and for survival. Here was different. Suddenly everybody was holding hands for the greater purpose and wanted to. (P7)
	Enablers to success	48	Having good flow and people managing the flow. People know what’s happening and making sure it’s smooth and people are in the right line and having the COVID marshals, really clearly defined roles so people know whose role is what, having the high-viz vests, that kind of thing. (P4)
	Logistics	37	They have a huge vaccine fridge there so we always knew that… it’s literally a three-minute walk across to their fridge where we could put it in, make sure that that’s maintained. (P1)
	Lessons learned	34	We were very open to anybody coming in and not discriminating against any particular person or group of people. But there were people queuing up who were clearly housed, wealthy, local residents. (P2.
	Barriers	27	There were still people confined to their flats and there were people that—I mean we immunized a lot of people with heavy addiction issues. But there are still those people that hardly leave their apartments. (P2)
	Flexibility	27	The rehabs, for example, they contacted us and said that they were coming down, and they had 12 or 14 people, so that we were prepared for that and we could make sure that we could fit them in. (P12)
	Promotion	15	We had these flyers… We sent them everywhere. We distributed them through all the homelessness interagency groups that we had. And we took them on patrol. (P4)
	Project dynamics	11	You’re dealing with a disparate group of people who are spread out all over the place, who are highly vulnerable. The more the merrier, spread the net wide, work together, we had a much better chance of getting to those people in greatest need by spreading the load across multiple services. (P9)
	Inconsistent information	9	In Australia, there was a lot of confusion about the eligibility and the eligibility criteria. (P2)

## Data Availability

The data presented in this study are available on request from the corresponding author.

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
