# Peer review of "Optimizing Access to the COVID-19 Vaccination for People Experiencing Homelessness"

_ijerph, 2022, doi:10.3390/ijerph192315686_

Round 1
Reviewer 1 Report
This is a well-written paper on an important aspect regarding access to health care. It addresses access to health care for the homeless, a vulnerable group that has significantly more difficult circumstances when it comes to accessing health care. By presenting COVID-19 vaccination and the establishment of a vaccine hub in Sydney it describes how homeless people are supported to receive COVID-19 vaccinations. The paper is clearly structured, and the supplementary material in particular is helpful.
However, there are some aspects that need improvement:
1. the research questions stated in the Materials and Methods section do not fully match the results. The discussion states that this is the first study to examine COVID-19 vaccine hesitancy among people experiencing or at risk of homelessness in Australia (line 410/411). This objective was not stated as a research question. Hence, the research questions need to be worded differently or supplemented.
2. line 32: what is meant by "culturally appropriate way"? It would be good to give an explanation.
3. line 63-65: it would be nice to add here why these groups of people have such a high risk of becoming homeless in Australia.
4. line 71: it would be good to add here whether this finding is independent of the duration of homelessness. If there are no correlations with duration of homelessness, it would be interesting to know why this is the case.
5. lines 79-86: in order to present the health problems of homeless people a little more clearly, it would be necessary to add some information also on the mental health situation of homeless people.
6. the research questions mentioned at the beginning of the methods section should be integrated at the end of the introduction.
7. table 1: it is noticeable that mainly questions were asked that presuppose a success of the vaccine hub. Such a question strategy influences the answers of the interviewees. There are no questions about what may not have gone well in the implementation of the vaccine hub. Only the last question is aimed at improvements that could be made. This seems to me to be a clear limitation of the study that also needs to be mentioned in the limitations.
8. line 158: how was it checked whether the participants of the survey were able to give informed consent to the study? This should definitely be described.
9. line 180: it is very surprising that only 49 participants could be recruited for the survey between September 23 and October 28, 2021. What were the problems regarding recruitment? Based on the numbers presented, it can be assumed that more than 20 vaccinations took place daily. Why then were fewer than 2 participants recruited each day? The problems should be addressed or explained why there were so few participants.
10. table 2: is misleading regarding the ages. Only people over 18 were surveyed, so it should read 18-24 and not 15-25.
11. point 3.3.3: it is pleasing to read that apparently also people could be reached who could not visit the hub. However, the question then arises as to whether the establishment of the hub is actually the effective strategy or the contacts with the homeless that could have been established by meeting homeless people in the hub. This should be discussed in the discussion. It is important to mention the percentage of homeless people who did not visit the hub but were driven to.
12. line 413-419: in my opinion, this section belongs in the result section.
13. the discussion is very short and should urgently be better organized in terms of the research questions. There is also a lack of critical commentary on whether it is better to serve the homeless through a hub or by driving to the homeless. What are the strengths of a hub? Why should this solution be chosen? Should a hub always be associated with outreach work? These questions should be addressed in far more detail.
Author Response
|
|
Reviewer One |
Response |
|
1 |
The research questions stated in the Materials and Methods section do not fully match the results. The discussion states that this is the first study to examine COVID-19 vaccine hesitancy among people experiencing or at risk of homelessness in Australia (line 410/411). This objective was not stated as a research question. Hence, the research questions need to be worded differently or supplemented. |
The third objective has been amended as follows: “Exploration of the perceptions of client’s receiving the COVID-19 vaccination regarding the Inner City COVID-19 Vaccine Hub model of care and its impact in relation to their access to vaccination, including vaccine hesitancy. (Line 141)
|
|
2 |
Line 32: what is meant by "culturally appropriate way"? It would be good to give an explanation.
|
The sentence has been amended as follows: “The urgency of the Australian COVID-19 national vaccination program requires an ongoing, integrated, and collaborative approach to ensure that people experiencing homelessness are supported to access COVID-19 vaccination in a culturally appropriate way that fosters a sense of safety by acknowledging peoples sense of cultural identity.” (Line 31-33) |
|
3 |
Line 63-65: it would be nice to add here why these groups of people have such a high risk of becoming homeless in Australia.
|
A sentence has been added as follows: “Issues that contribute to homelessness, or escalate during a period of homelessness, are disproportionately experienced by Aboriginal and Torres Strait Islander people, including chronic disease, mental health concerns, social disadvantage, domestic violence and long-standing inequity in access to health and social services” (Lines 67-70) |
|
4 |
Line 71: it would be good to add here whether this finding is independent of the duration of homelessness. If there are no correlations with duration of homelessness, it would be interesting to know why this is the case.
|
The finding is independent of length of homelessness. Duration of homelessness was not included as a variable in the study. The authors used emergency department data to conduct the retrospective study, and the length of homelessness would not necessarily have been available to be included as an independent variable. The type of homelessness (primary, secondary, tertiary) was available as were the demographic details of participants (Seastres et al., 2020). |
|
5 |
Lines 79-86: in order to present the health problems of homeless people a little more clearly, it would be necessary to add some information also on the mental health situation of homeless people.
|
Mental health is identified as a risk factor for homelessness at line 83.
|
|
6 |
The research questions mentioned at the beginning of the methods section should be integrated at the end of the introduction. |
The research questions have been moved to the end of the introduction (Lines 132-141) |
|
7 |
Table 1: it is noticeable that mainly questions were asked that presuppose a success of the vaccine hub. Such a question strategy influences the answers of the interviewees. There are no questions about what may not have gone well in the implementation of the vaccine hub. Only the last question is aimed at improvements that could be made. This seems to me to be a clear limitation of the study that also needs to be mentioned in the limitations.
|
The following sentence has been added to the limitations section: “Only people who received the vaccine were invited to complete the survey, which means the survey responses somewhat presupposed the success of the Vaccine Hub. The views of people choosing not to engage with the Vaccine Hub are absent from the findings. There may have been some important feedback from those who did not engage with the Vaccine Hub.” Lines 482-485) |
|
8 |
Line 158: how was it checked whether the participants of the survey were able to give informed consent to the study? This should definitely be described.
|
The following has been added: “The researcher was accompanied by a Peer Support Worker with lived experience of homelessness and/or an Aboriginal Health Worker. The researcher explained the purpose of the survey and provided a participant information sheet, which was read to participants as required. Participants were invited to ask any questions. Consent to participate was provided verbally, and was electronically recorded on the survey tool by the researcher.” (Line 169-171) |
|
9 |
Line 180: it is very surprising that only 49 participants could be recruited for the survey between September 23 and October 28, 2021. What were the problems regarding recruitment? Based on the numbers presented, it can be assumed that more than 20 vaccinations took place daily. Why then were fewer than 2 participants recruited each day? The problems should be addressed or explained why there were so few participants.
|
The following has been added: “The survey was conducted between 23rd September and 28th October 2021 at each weekly Vaccine Hub session. The research team member collecting the data was sometimes required to support the administration of vaccines and was often available to administer the survey for half day periods. The frequency of data collection is reflected in the number of surveys completed (n=49).”
|
|
10 |
Table 2: is misleading regarding the ages. Only people over 18 were surveyed, so it should read 18-24 and not 15-25.
|
This error has been amended (Page 5) |
|
11 |
A) Point 3.3.3: it is pleasing to read that apparently also people could be reached who could not visit the hub. However, the question then arises as to whether the establishment of the hub is actually the effective strategy or the contacts with the homeless that could have been established by meeting homeless people in the hub. This should be discussed in the discussion. B) It is important to mention the percentage of homeless people who did not visit the hub but were driven to. |
A) The discussion reflects these findings in lines 430-432 which discuss the importance of person-centred and trauma informed approaches. Lines 433-444 also discuss the need for a tailored and targeted approaches for optimizing access to the vaccine for people experiencing homelessness. We have not altered the discussion as we feel the establishment of the hub, associated outreach and person-centred approaches are all effective strategies.
We do not have this information; gathering this data was beyond the scope of the study. |
|
12 |
line 413-419: in my opinion, this section belongs in the result section. |
This section is an interpretation of the findings and suggests that the Vaccination Hub met its intended purpose. Therefore we have left in the discussion. (Lines 426-432) |
|
13 |
the discussion is very short and should urgently be better organized in terms of the research questions. There is also a lack of critical commentary on whether it is better to serve the homeless through a hub or by driving to the homeless. What are the strengths of a hub? Why should this solution be chosen? Should a hub always be associated with outreach work? These questions should be addressed in far more detail.
|
The discussion has been expanded to include discussion of whether outreach or clinic based vaccination is optimum. |
Reviewer 2 Report
Thank you for letting me review this very important piece of work. It is a fantastic initiative, and well needed.
I offer some suggestions to strengthen the paper.
Title:
· As the aim of the project is to ultimately create the blueprint document, I am not sure if you want to represent this in the title?
Abstract:
· You mention 4305 vaccines were administered. This is not written anywhere in the main text. Please remove, or add into the main results.
Introduction:
· Lines 31-35 – It would be good to understand what these numbers mean in the context of the Australian public.
· Lives 38-39 – for the international reader it would be good to introduce the Torres Strait Islander people, and their context
· Lines 45-49 – unsure if this would be better at the end of the introduction, after the background of the situation has been laid out
· Lines 82 – full stop needed after reference 12.
· Lines 97-98 – can those who are homeless or at risk of homelessness not access GP care?
· Line 107 – ‘a common approach to messaging.’ – How was the service publicized to eligible participants? Were eligible participants supported to get to the centre?
· Section 1.4 may be better placed in the methodology as a programme design section, although there may be a reason why you have it in the introduction
· Lines 118-121. No mention of aim here. You should echo what is written in the abstract here. Currently they are different.
Methods:
· Section 2.1 and 2.2 headings are missing. You start at 2.3. Please correct.
· 2.3.1 – it would be good to know the roles or backgrounds of those who were interviewed.
· For the interviews you may wish to include a completed COREQ checklist in an appendix, although this is not essential
· 2.3.2 – was a participant information leaflet provided to participants. During the survey period how were the 45 respondents identified? Were all eligible respondents asked if they wanted to participate, or was the survey only conducted at certain time points during the month of data capture? More information is needed.
· Lines 169-174- this should be a separate section related to ‘data analysis’
· Although ethics is mentioned in the additional information at the end of the article, it would be good to acknowledge it in the full text
· For results listed in the text from lines 190-216 please add n numbers along with the percentage score.
· Line 218 – ‘no significant associations’ – please add P values.
· 3.3. I would suggest only use quotes in one area – either in the table or in the text. Having different quotes in different places made it hard to follow. In table 4, adding N would add an element of content analysis, and I feel these are not required. However, if you want to include them, it may be better to put in order from most to least to more easily see the codes and their importance in the theme
· Lines 351-357 – this is context regarding the service, not a result, so would be better in the introduction or in the discussion
Discussion and conclusion:
· Line 404 – COVI should read COVID
· Line 424 – UK should be expanded to United Kingdom (UK)
· Lines 457-461 – great supplementary file, but more specifics about how the blueprint might be helpful and which specific elements were included would be good to be clear for the reader
· In conclusions is there any future work, or how (apart from this paper!) you will be sharing best practice or publicizing or piloting the blueprint created?
Thank you again for letting me review this very important piece of work
Author Response
|
14 |
As The aim of the project is to ultimately create the blueprint document, I am not sure if you want to represent this in the title?
|
From our perspective, the ultimate purpose is to optimise access to the COVID-19 vaccination. Developing an evidence-based model of care underpinned this purpose. For this reason we have kept the title unchanged. |
|
15 |
· You mention 4305 vaccines were administered. This is not written anywhere in the main text. Please remove, or add into the main results.
|
Thank you. This information has been added to the body of the paper (lines 123-124). |
|
16 |
· Lines 31-35 – It would be good to understand what these numbers mean in the context of the Australian public.
|
Percentages of the Australian public have been added to this section. (Lines 34-36) |
|
17 |
· Lines 38-39 – for the international reader it would be good to introduce the Torres Strait Islander people, and their context
|
A foot note has been added to page 2 to explain this context. It reads: “The Torres Strait Islands stretch between the coast of Northern Australia and Papua New Guinea, with Indigenous populations that are distinct from those of Australia’s mainland.” |
|
18 |
L Lines 45-49 – unsure if this would be better at the end of the introduction, after the background of the situation has been laid out. |
The aims/objectives have been moved to the end of the introduction (lines 132-141). |
|
19 |
Lines 82 – full stop needed after reference 12 |
This full stop has been added. (Line 88) |
|
20 |
Lines 97-98 – can those who are homeless or at risk of homelessness not access GP care?
|
People experiencing homelessness can access GP care, but unfortunately they tend not to in Australia. Instead they access care via an emergency department. We have added the words ‘General Practitioner’ to line 86 to make this clearer. |
|
21 |
· Line 107 – ‘a common approach to messaging.’ – How was the service publicized to eligible participants? Were eligible participants supported to get to the centre?
|
This information is included in the Blueprint (Supplementary File). We have added the following sentence at Line 115: “The Vaccine Hub was promoted using flyers, which were distributed through existing networks.” |
|
22 |
Se Section 1.4 may be better placed in the methodology as a programme design section, although there may be a reason why you have it in the introduction
|
This section is placed in the introduction to ensure the reader has context for the establishment of the Vaccination Hub. We have not moved it to a ‘programme design’ section. |
|
23 |
Lines 118-121. No mention of aim here. You should echo what is written in the abstract here. Currently they are different. |
The aim has been updated here to reflect that of the abstract. (Lines 126-127) |
|
24 |
Section 2.1 and 2.2 headings are missing. You start at 2.3. Please correct.
|
The heading numbers have been corrected as advised. |
|
|
· 2.3.1 – It would be good to know the roles or backgrounds of those who were interviewed.
|
The following has been added to provide further detail, without revealing the specific roles, as this would identify the individual participants. Line 155: “The roles of participants varied and included nurses, nurse unit managers, health service managers and those in managerial and executive roles in member organisations of the Intersectoral Homelessness Health Alliance.” |
|
25 |
For the interviews you may wish to include a completed COREQ checklist in an appendix, although this is not essential |
The checklist has not been added to the appendix |
|
26 |
2 2.3.2 – was a participant information leaflet provided to participants. During the survey period how were the 45 respondents identified? Were all eligible respondents asked if they wanted to participate, or was the survey only conducted at certain time points during the month of data capture? More information is needed. |
Further information has been provided at Lines 174-179 and at Lines 190-194. |
|
27 |
Lines 169-174- this should be a separate section related to ‘data analysis’ |
A new sub-heading ‘data analysis’ has been added (line 180). |
|
28 |
Although ethics is mentioned in the additional information at the end of the article, it would be good to acknowledge it in the full text |
A line has been added to page 4 (lines 187-188) |
|
29 |
For results listed in the text from lines 190-216 please add n numbers along with the percentage score. |
N numbers have been added to this section of text alongside percentage scores (Lines 205-229) |
|
30 |
Line 218 – ‘no significant associations’ – please add P values. |
The Chi-Square analysis was undertaken with each of the demographics (age, gender, Aboriginal and or Torres Strait Islander) therefore it is difficult to report each of the p values. For this reason we have chosen not to report them in the paper, and have copied them into a table for your review below. |
|
31 |
· 3.3. I would suggest only use quotes in one area – either in the table or in the text. Having different quotes in different places made it hard to follow. In table 4, adding N would add an element of content analysis, and I feel these are not required. However, if you want to include them, it may be better to put in order from most to least to more easily see the codes and their importance in the theme. |
We have decided to keep table 4 in the article. We have reconfigured the table with codes from most frequent to least frequent within each theme as suggested (Page 10) |
|
32 |
Lines 351-357 – this is context regarding the service, not a result, so would be better in the introduction or in the discussion. |
This context relates to the participant’s motivations for establishing the hub and the logistics of doing so, which were discussed during the interviews. Therefore, we have decided for this information to remain in the results section. |
|
33 |
Line 404 – COVI should read COVID |
This has been amended (Line 418) |
|
34 |
Line 424 – UK should be expanded to United Kingdom (UK) |
This has been amended (Line 437) |
|
35 |
Lines 457-461 – great supplementary file, but more specifics about how the blueprint might be helpful and which specific elements were included would be good to be clear for the reader |
We have added to this sentence as follows: “It is hoped that the Vaccination Hub Blueprint, including philosophy of care, collaborative practices, resources, logistics and lessons learned, will be helpful to other services that provide vaccination to people experiencing, or at risk of homelessness, and to those experiencing any form of social marginalization.” Lines 472-473). |
|
36 |
In Conclusions is there any future work, or how (apart from this paper!) you will be sharing best practice or publicizing or piloting the blueprint created? |
The Blueprint has already been shared with our existing networks. We have no other plans to share this work, other than this publication. |
Round 2
Reviewer 1 Report
The authors have been able to improve the manuscript considerably, it should be accepted in the present form.